# A Combination of Laboratory Testing, RCE, and Corrosion Loop for Inhibitor Selection

**Gustavo Luis Bianchi** [1,2,*], **Verónica Acosta** [2,3] **and Carlos Seijas** [1]

1   Energy and Environmental Innovation Group, Faculty of Engineering, Malvinas Institute,
    National University of La Plata, La Plata B1900, Argentina
2   National Scientific and Technical Research Council (CONICET), Buenos Aires C1425FQB, Argentina
3   Faculty of Engineering, National University of Patagonia, San Juan Bosco,
    Comodoro Rivadavia U9000, Argentina
*   Correspondence: gustavo.bianchi@ing.unlp.edu.ar

**Featured Application: Specific applications for the oil and gas industry. Testing of corrosion inhibitors to select the one with the best performance simulating field conditions.**

**Abstract:** Corrosion inhibitors are evaluated in the oil industry with electrochemical tests of resistance to linear polarization with rotating cylinders following ASTM G170 and NACE 3T199 standards. With these tests, we can determine the corrosion rate (CR) and efficiency of corrosion inhibitors. In this work, a corrosion test protocol used by hydrocarbon-producing companies for the testing of corrosion inhibitors was used. This protocol consists of a 1045 carbon steel working electrode in a NACE solution composed of 9.62% NaCl, 0.45% $CaCl_2$, 0.19% $MgCl_2$, and 89.74% $H_2O$, at a temperature of 65 °C and saturated with $CO_2$. Each inhibitor tested was subjected to a series of 6000-4000-2000-4000-6000 rpm tests using rotating cylinder electrodes (RCEs). These electrochemical studies were carried out with the rotating cylinder to evaluate the ability of the inhibitor to prevent the corrosion of carbon steel in the presence of a centrifugal force. In our opinion, this test does not provide corrosion engineers with enough information to be used as a predictive tool, since what is obtained is the CR in a very short testing time. This document proposes the use of two more appropriate test methodologies, the rotating cylinder electrode (RCE) and the flow loop (FL), to evaluate the performance of the corrosion inhibitor. For the FL, the selected flow rate was 1.2 m/s, the same rate that fluids have in oil company pipelines installed in Neuquén, Argentina. Firstly, according to the company's protocol, inhibitors are required to have an efficiency greater than or equal to 90% in RCE tests; therefore, inhibitors that meet these requirements were tested in the FL test. Unlike the RCE test, the FL test represents the experimental conditions of the laboratory that are closest to reality, for the evaluation of the performance of the inhibitors used in the pipelines of the oil and gas industry. FL tests have several problems involving corrosion, erosion, abrasion, biphasic fluids, the time it takes for the inhibitor to become effective, and the duration of its effectiveness.

**Keywords:** rotating cylinder electrodes; corrosion loop; resistance to linear polarization

## 1. Introduction

In the oil and gas industry, the severity of $CO_2$ corrosion (or sweet corrosion) of pipelines is affected by crossed reactions and the interaction of different factors, including environmental, metallurgical, and physical parameters. The main environmental variables that affect $CO_2$ corrosion are temperature, pressure, flow condition, type of material, and salt concentration. Moreover, impurities in the surrounding flow may significantly impact the corrosion of carbon steel [1,2]. The main concern of $CO_2$ corrosion is that it may lead to equipment failure, especially in the tubing/casing of the production and injection wells, as well as pipelines, thus interrupting the oil/gas production [3,4]. Even the slightest change in one of these parameters can considerably impact the corrosion rate, given the

property changes in the thin layer of corrosion products accumulated on the surface of the steel [5,6]. When corrosion products are not on the surface of the steel, high rates of corrosion of several millimeters per year can take place. Corrosion rates can be substantially reduced under the conditions in which iron carbonate ($FeCO_3$) can precipitate on the steel surface and create a dense and protective corrosion product film. This occurs more easily at high temperatures or high pH in the water phase [6,7]. High temperature increases the rate of reaction, which produces a protective corrosion product layer, thus reducing corrosion rates, while high pH reduces the descaling of the corrosion product layer, which yields a low corrosion rate [3,8]. Dry $CO_2$ is not corrosive under the temperatures found in oil and gas production. It must dissolve in a liquid phase to cause an electrochemical reaction between the steel and the liquid phase in contact. $CO_2$ is soluble in water and brine. However, it should be pointed out that it has a solubility rate similar to that of hydrocarbon, both in the gaseous and liquid phases. Thus, for a mixed-phase system, the presence of the hydrocarbon phase may provide $CO_2$ deposition for the partition in the liquid phase. $CO_2$ is usually found in the produced fluids. Basically, $CO_2$ corrosion is an electrochemical process consisting of several steps that occur between the corrosive species that result from $CO_2$ dissolution and the soluble phases of steel. The corrosion of tubing in a $CO_2$ environment has been widely studied under various conditions to understand its corrosion mechanism and protective methods [9–27]. Using chemical treatment to prevent iron and steel from $CO_2$ corrosion is one of the most cost-effective ways [28–30]. Organic molecules are one of the most popular, efficient, and practical corrosion inhibitors, and most studies demonstrate that organic compounds, preferentially those containing nitrogen (N), sulfur (S), and oxygen (O), have high inhibitory efficiency [31–35].

D. Harrop et al. [36] specifically compared essays on corrosion inhibitors carried out in a flow loop with a rotating cylinder. The obtained corrosion rates revealed significant differences depending on time, given that they were tested at 1 bar of $CO_2$, 50 °C, pH 5.6, and 20 Pa shear stress. These differences exist because of the different shear stress generated in FL and RCE tests on the surface of each tested sample. Shear stress equations for both FL and RCE tests are shown below [37–41].

$$\text{RCE } \sigma_c = 0.079 \text{ Re}^{-0.3} dv^2 \tag{1}$$

$$\text{FL } \sigma_p = 0.039 \text{ Re}^{-0.25} dv^2 \tag{2}$$

where $\sigma_c$ = turbulent shear stress (cylinder), Re = Reynolds number, d = fluid density, $\sigma_p$ = turbulent shear stress (pipe), and v = fluid velocity.

Nowadays, the efficiency of corrosion inhibitors under certain flow conditions can be assessed using RCEs in accordance with ASTM [42–44].

Corrosion engineers require a predictive tool that is not over-conservative and that will not compromise safety. When designing a facility or pipeline, requirements such as service life, corrosion allowance, and the efficacy of inhibitor treatment dictate the quantification of the corrosion rate. For this reason, the aim of this work is the implementation of a methodology for the evaluation of corrosion inhibitors combining RCE and FL methods, to select the parameters for the inhibitor's best performance.

## 2. Experimental Development

Corrosion inhibitors called Inhibitor 1, Inhibitor 2, Inhibitor 3, and Inhibitor 4 of an oil company from Argentina were tested at a 150 ppm concentration. A NACE solution was used, as was requested by the company according to its protocol, composed of the following salts: 9.62% NaCl; 0.45% $CaCl_2$; 0.19% $MgCl_2$; and 89.74% $H_2O$. All reagents are of analytical grade, and distilled water was used for its dissolution. For all tests, the working temperature was 65 °C, saturated with $CO_2$ 99.995%. At 25 °C, the $CO_2$-saturated NACE solution's pH was 5.27.

*2.1. Electrochemical Tests of Resistance to Linear Polarization with Rotating Cylinder*

Tests were carried out at a laboratory scale to assess the efficiency of inhibitors on the corrosion rate, using linear polarization resistance (LPR) in dynamic conditions in an electrochemical cell.

In the linear polarization tests, the Metrohm AUTOLAB potentiostat/galvanostat was used, along with its corresponding software, NOVA 2.1.

The electrochemical cell used has three electrodes: the reference electrode, which was AgCl/Ag in a KCl 3M solution brought to closer contact with the working electrode using a Luggin capillary, thus lowering the ohm resistance between the reference electrode and the working electrode; the working electrode, a 1045 carbon steel cylinder with a 2.47 cm$^2$ exposed area; finally, a graphite counter-electrode was used. All electrodes were submerged in the NACE solution.

To carry out the tests in dynamic conditions, a rotating disk electrode was used, consisting of a rotating rod. The rotation working speed was, in accordance with the oil company's protocol, a series of 6000-4000-2000-4000-6000 rpm for each studied inhibitor. The working electrode rotated in the core of the NACE solution at a working temperature of 65 °C with $CO_2$ bubbling.

To assess the efficiency of the inhibitors (ASTM G185-06), the instantaneous corrosion rate was measured using the LPR method at the aforementioned speeds, with constant $CO_2$ bubbling at 65 °C. After one-hour gas saturation, the open-loop potential was measured for one hour, with and without a corrosion inhibitor, followed by a polarization resistance test between −20 mV and +20 mV of the open-loop potential.

To calculate the efficiency, Equation (3) is used, where CR is the corrosion rate:

$$Efficiency = \frac{CR\ without\ inhibitor - CR\ with\ inhibitor}{CR\ without\ inhibitor} * 100 \tag{3}$$

*2.2. Electrochemical Tests in Corrosion Loop*

For the dynamic testing, FL was used (Figure 1A). It has a storage capacity of 18 liters. In the loop pipelines, two 1045 carbon steel corrosion coupons were placed, labeled as C1 (the upper area of the pipeline) with a medium-exposed area of 56.5488 mm$^2$ and as C2 (lower area of the pipeline) with a medium-exposed area of 64.8168 mm$^2$. To avoid crevice corrosion formation in the coupons, they were carefully sealed with neutral cure silicone, as shown in Figure 1B. Before mounting the coupons, they were glass-polished and weighed with an AUW 220D weighing scale (e = 1 mg; d = 0.1 mg/0.01 mg).

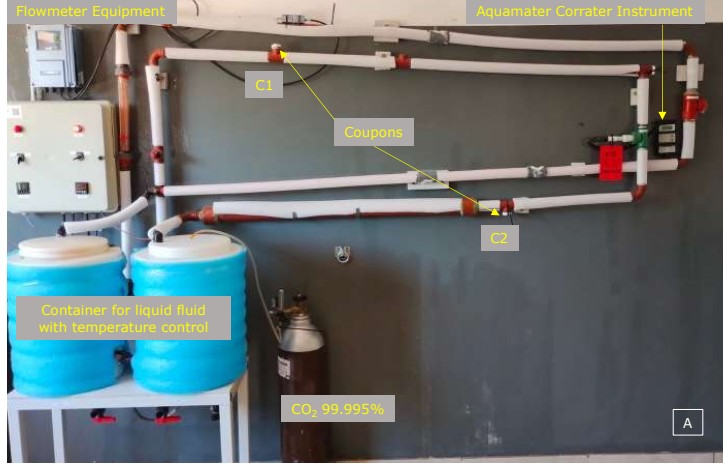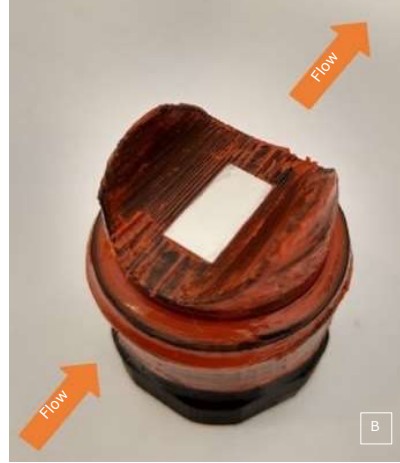

**Figure 1.** (**A**) Corrosion loop and its components; (**B**) coupon.

## 3. Results and Discussion

Table 1 and Figure 2 show the test results for each assessed inhibitor. The target, a tested graduated cylinder without an inhibitor, which was used to obtain the inhibitors' efficiency, had a corrosion rate (CR) of 19.21 mm/year with a corrosion potential (Ec) of −683 mV and a corrosion current (CC) of 2.5 mA.

**Table 1.** Polarization resistance essays in a rotating cylinder in a NACE solution. Corrosion rate (CR) and inhibitor's efficiency with respect to the general corrosion.

| RPM | Inhibitor 1 | Inhibitor 2 | Inhibitor 3 | Inhibitor 4 |
|---|---|---|---|---|
| 6000 | CR = 1.71 mm/year<br>$E_c$ = −619 mV<br>CC = 0.38 mA<br>Efficiency = 91.10% | CR = 7.89 mm/year<br>$E_c$ = −552 mV<br>CC = 1.5 mA<br>Efficiency = 58.93% | CR = 6.62 mm/year<br>$E_c$ = −535 mV<br>CC = 1.33 mA<br>Efficiency = 65.54% | CR = 1.8 mm/year<br>$E_c$ = −622 mV<br>CC = 0.29 mA<br>Efficiency = 90.63% |
| 4000 | CR = 1.27 mm/year<br>$E_c$ = −616 mV<br>CC = 0.27 mA<br>Efficiency = 93.39% | CR = 4.57 mm/year<br>$E_c$ = −576 mV<br>CC = 0.97 mA<br>Efficiency = 76.21% | CR = 6.18 mm/year<br>$E_c$ = −613 mV<br>CC = 1.30 mA<br>Efficiency = 67.83% | CR = 1.49 mm/year<br>$E_c$ = −612 mV<br>CC = 0.31 mA<br>Efficiency = 92.24% |
| 2000 | CR = 1.23 mm/year<br>$E_c$ = −596 mV<br>CC = 0.26 mA<br>Efficiency = 93.60% | CR = 4.78 mm/year<br>$E_c$ = −578 mV<br>CC = 1.22 mA<br>Efficiency = 75.12% | CR = 1.8 mm/year<br>$E_c$ = −600 mV<br>CC = 0.29 mA<br>Efficiency = 90.63% | CR = 1.62 mm/year<br>$E_c$ = −603 mV<br>CC = 0.38 mA<br>Efficiency = 89.85% |
| 4000 | CR = 1.43 mm/year<br>$E_c$ = −596 mV<br>CC = 0.30 mA<br>Efficiency = 92.56% | CR = 4.08 mm/year<br>$E_c$ = −578 mV<br>CC = 1.12 mA<br>Efficiency = 78.76% | CR = 7.72 mm/year<br>$E_c$ = −550 mV<br>CC = 1.60 mA<br>Efficiency = 59.81% | CR = 1.95 mm/year<br>$E_c$ = −602 mV<br>CC = 0.41 mA<br>Efficiency = 89.85% |
| 6000 | CR = 1.32 mm/year<br>$E_c$ = −599 mV<br>CC = 0.28 mA<br>Efficiency = 93.13% | CR = 6.18 mm/year<br>$E_c$ = −568 mV<br>CC = 1.30 mA<br>Efficiency = 67.83% | CR = 7.72 mm/year<br>$E_c$ = −555 mV<br>CC = 1.61 mA<br>Efficiency = 59.81% | CR = 1.77 mm/year<br>$E_c$ = −589 mV<br>CC = 0.37 mA<br>Efficiency = 90.79% |

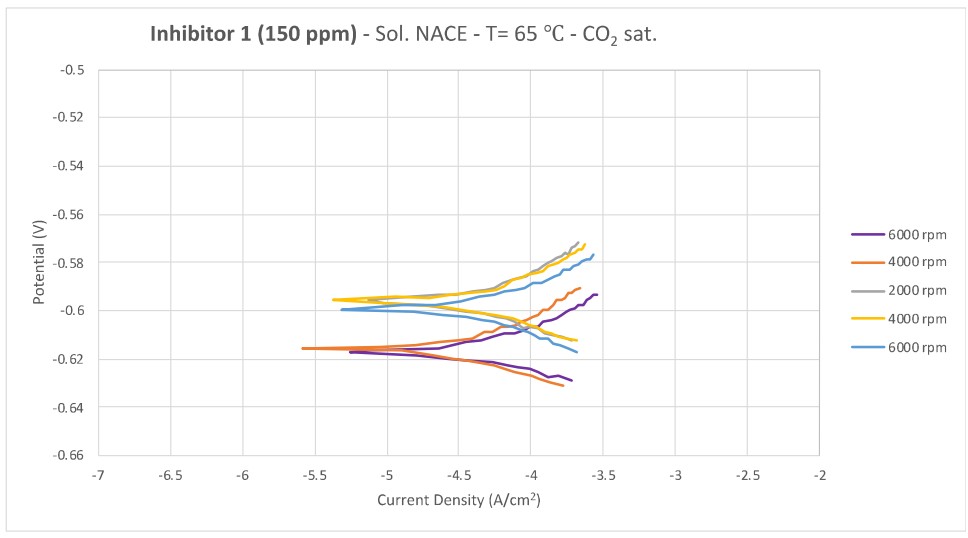

**Figure 2.** *Cont.*

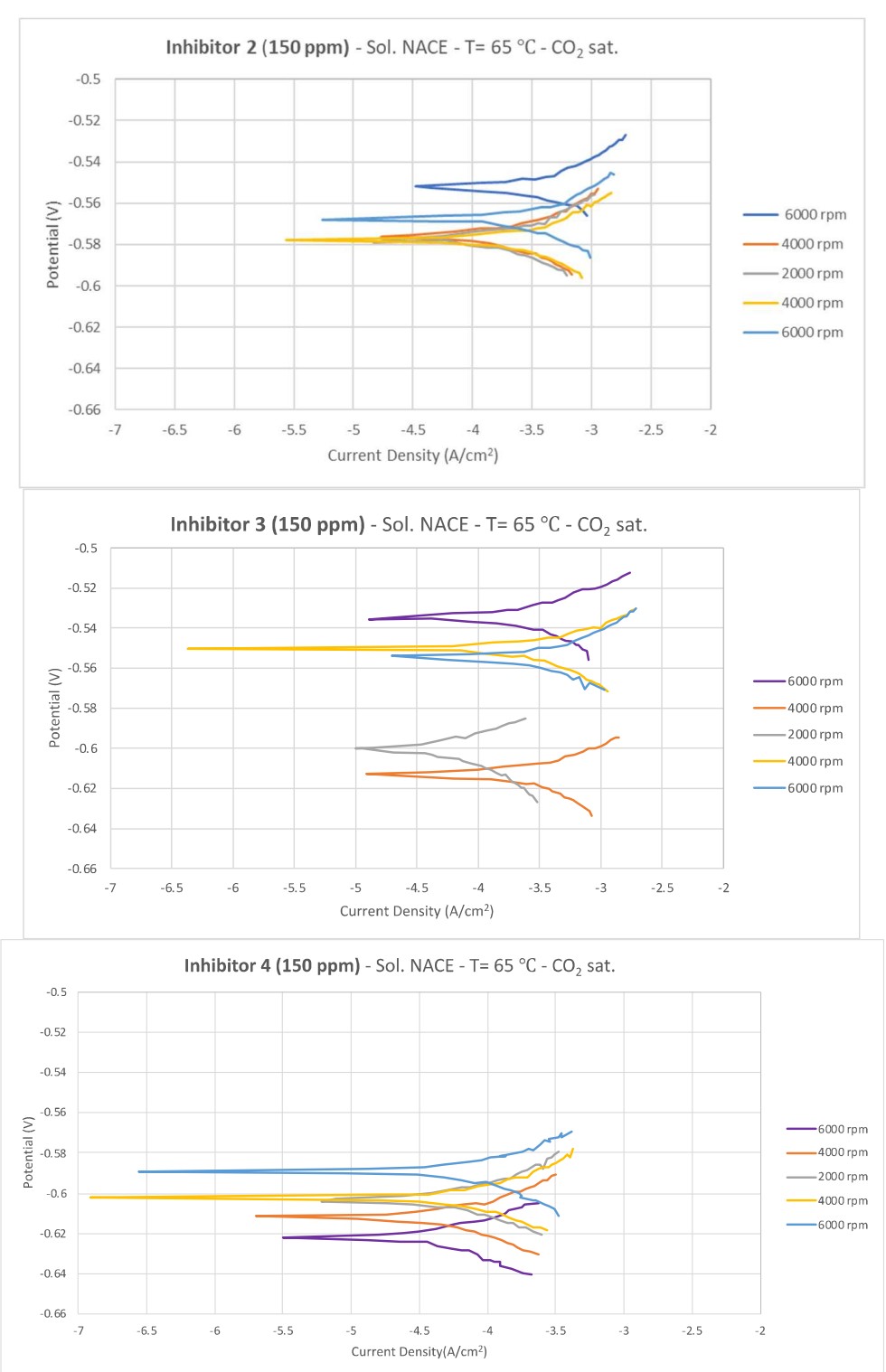

**Figure 2.** Curves obtained through polarization resistance to different rpm of the corrosion inhibitors.

Out of the four inhibitors that were tested at a 150 ppm concentration, Inhibitor 1 was the one that had an efficiency equal to or greater than 90%. The rest of the corrosion inhibitors did not reach a 90% efficiency in any of the tests with the rotating cylinder. The oil company considers that an inhibitor passes the test if its corrosion efficiency is equal to or greater than 90%.

Fractographies at 20× of the carbon steels tested with the corrosion inhibitors are shown in Figure 3. They were assessed considering the criterion of Smith et al. [45], which is shown in Table 2, as exclusively requested by the oil company.

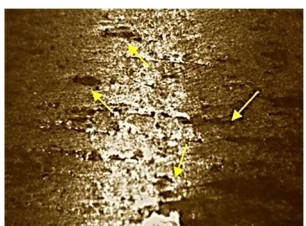

White – No Inhibitor – Smith Criterion = 7

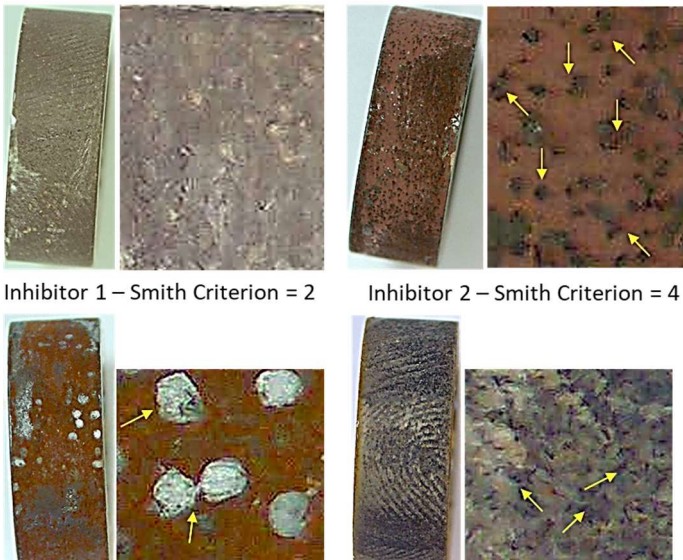

Inhibitor 1 – Smith Criterion = 2

Inhibitor 2 – Smith Criterion = 4

Inhibitor 3 – Smith Criterion = 5

Inhibitor 4 – Smith Criterion = 4

**Figure 3.** Fractographies of the tested graduated cylinders (20×). The yellow arrows show the presence of pitting according to Smith's criterion.

**Table 2.** Smith's criterion for assessing corrosion levels.

| Rank | Description of Pitting |
| --- | --- |
| 0 | No pits. Surface same as for original untreated coupon. |
| 1 | Intergranular corrosion on cut edge of coupon, giving a sintered effect; no pits on major surfaces. |
| 2 | Small, shallow pits on cut edges; no pits on major surfaces. |
| 3 | Scattered, very shallow pinpoint pits, less than 25 pits on either surface—that is, front or back. |
| 4 | More than 25 pits of Rank 3 on either surface. |
| 5 | Ten or fewer pits, 1/332 to 1/16 in. diameter and 1/64 to 1/32 in. deep. |
| 6 | Between 11 and 25 pits of Rank 5. |
| 7 | More than 25 pits of Rank 5. |
| 8 | Pits larger than 1/16 in. but less than 1/8 in. in diameter, greater than 1/32 in. deep, 100 or fewer in number. |
| 9 | Any pitting more severe than Rank 8. |

Inhibitor 1 exhibited general corrosion, but no major pits were observed; thus, it is considered to be Rank 2, according to Smith, whereas in the rest of the tested inhibitors, the surface of the carbon steel revealed abundant pitting.

Tests with the NACE solution (Table 3 and Figures 4 and 5) were carried out at a 65 °C working temperature, saturated with $CO_2$ 99.995%, at a flow speed of 1.2 m/s equal to an angular velocity of 1100 rpm [40]. The chosen flow velocity was the same as that of the pipeline in the company's oil field. A Cosasco Aquamate Corrater Instrument was used to measure both the electrochemical LPR and the weight-loss corrosion rate. An electrochemical current noise (ECN) form, known as imbalance (unbalance or pitting tendency) was also monitored among the electrodes. This is a useful qualitative instability signal on the surface of the material that coincides with pitting or localized corrosion. Figure 4 shows the corrosion rate and the imbalance rate (pitting tendency), both depending on time for each tested corrosion inhibitor. Inhibitors were added to the solution only once, with an individual concentration of 150 ppm, equal to the concentration used for the RCE tests.

**Table 3.** Measurements of corrosion rate relative to weight loss and LPR.

| Corrosion Inhibitor | 1 | 2 | 3 | 4 |
|---|---|---|---|---|
| C1 CR (mm/years) | 2.98 | 3.06 | 4.56 | 9.05 |
| C2 CR (mm/years) | 2.49 | 2.53 | 3.85 | 7.13 |
| LPR CR (mm/years) | 2.32 | 2.69 | 6.07 | 8.58 |

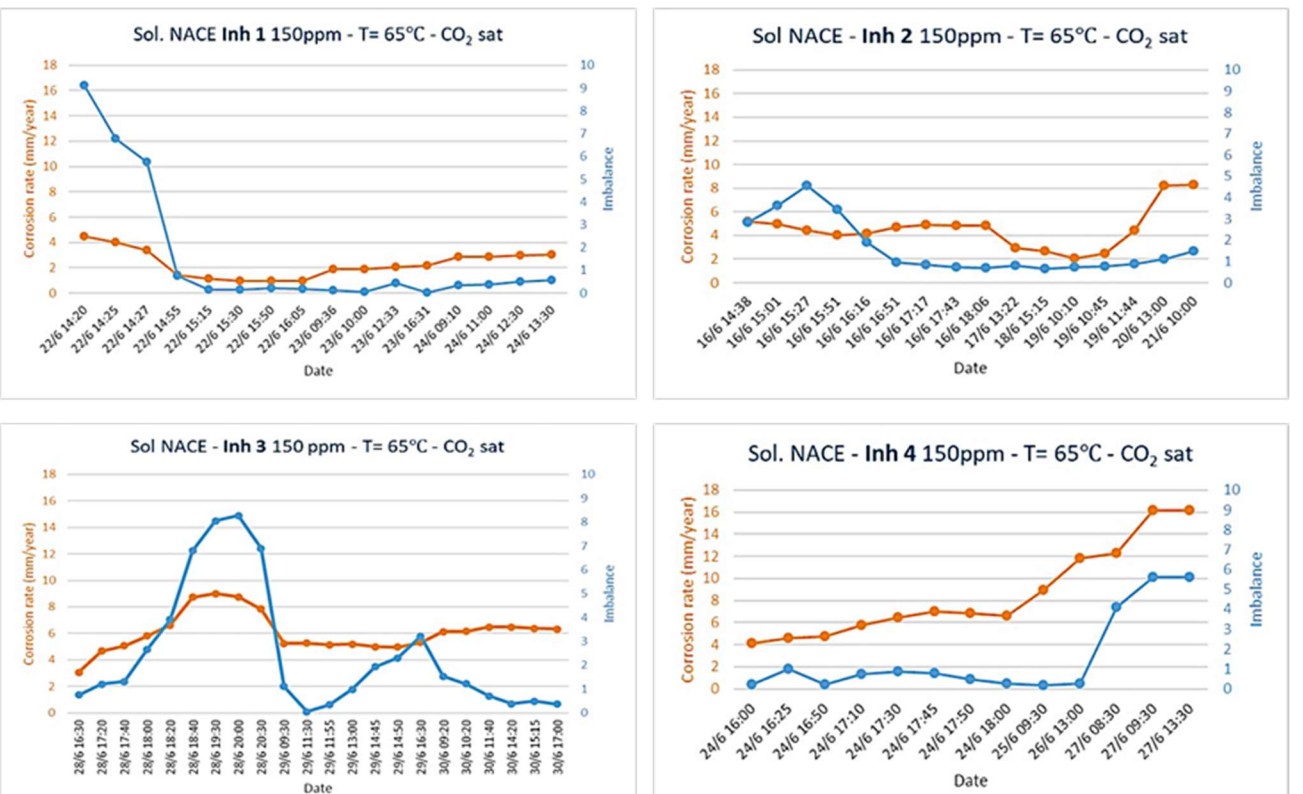

**Figure 4.** Curves obtained from each test. The *x*-axis shows the period of time, whereas the left *y*-axis shows the CR and the right *y*-axis shows the imbalance values.

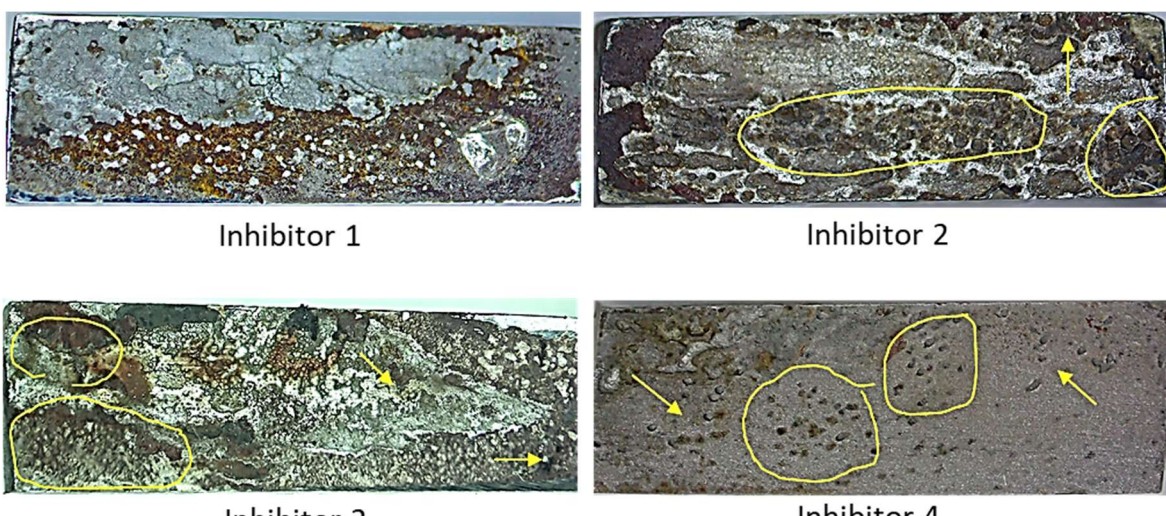

**Figure 5.** Pictures of the tested samples for each inhibitor. The circles and yellow arrows show the presence of localized corrosion (20×).

Electrochemical studies with rotating cylinders assess inhibitors' capacity to prevent carbon steel corrosion under a centrifugal force. They are tools that are commonly used to assess corrosion inhibitors' performance in oil fields [46–51].

As shown in Table 1 and Figure 2, at a constant inhibitor concentration (150 ppm) and under the aforementioned work conditions, the performance of the corrosion inhibitor (inhibition efficiency) depends on the working electrode rotation speed (shear stress). In most of the cases we studied, the higher the rpm, the higher the corrosion current (CC); that is, the higher the general corrosion rate, the lower the inhibitor's efficiency [52]. María Elena Olvera-Martínez et al. [52] obtained similar results when working with RCE at rates of 0, 100, 1000, 2000, 3000, 4000, 5000, and 6000 rpm, and using API 5L X52 steel cylindric samples as working electrodes in 5% NaCl solutions, at 60 °C, with saturated $CO_2$, and at different concentrations of the used inhibitor.

As can be seen in Figure 3, in this work, Inhibitor 1 was the one with an efficiency of over 90%, thus having a Rank 2 general corrosion, according to Smith's criterion.

We cannot speak of the efficiency of an inhibitor if there is localized corrosion. Efficiency against corrosion is defined as efficiency against general corrosion. If there is a localized attack, it is not possible to have efficiency values, given that there is inefficiency; that is, an inhibitor is not efficient if the material undergoes a localized attack (pitting, stress corrosion cracking, corrosion fatigue, etc.). This information is crucial for the corrosion engineer, given that an inhibitor that produces pitting endangers the security of the oil field's facilities. On the other hand, the fluid flow regime is one of the parameters that have a great effect on the surface rate. The type of flow regime affects the thickness and velocity of the water layer flowing up the tubing, and it defines the shear stress and mass transfer at the steel surface [53–55].

Flow loop tests are an important complement because they represent the closest experimental laboratory condition to assess the performance of those inhibitors that are used in the oil and gas industry. The selected flow rate, 1.2 m/s, is the same as that of the pipeline fluids of oil companies in Neuquén, Argentina. FL tests evince problems that involve corrosion, erosion, abrasion, two-phase fluids, the time the inhibitor takes to coat the graduated cylinders, the period during which the inhibitor works, etc. Table 3 shows the differences between the CR relative to the weight loss of corrosion coupons C1 and C2. The CR values found in corrosion coupon C1 were higher than those found in coupon C2. This is because C1 was on the upper part of the pipeline (Figure 3), where it came into contact with two-phase fluids ($CO_2$ gas + NACE solution), whereas corrosion coupon C2 was on the lower part of the pipeline, where it was in contact with the NACE solution +

dissolved $CO_2$. Table 3 shows the CR relative to the mean LPR of each inhibitor obtained with the Aquamate Corrater Instrument. These were within the CR value range obtained for each weight-loss coupon.

When assessing corrosion inhibitors, O. Gómez et al. [56] applied NACE G 170 in an FL test using a liquid solution containing pressurized $CO_2$. With the FL equipment they used, they determined the instantaneous corrosion rate with RLP and Tafel polarization curve (TPC) measurements, as well as the efficiency against the corrosion of the inhibitors, which was tested statically and dynamically at a 0.5 m/s flow rate.

We can obtain more information on this study by looking at Figure 6 (analysis of the first graphic of Figure 4), which corresponds to Inhibitor 1, where we can see different zones in which the inhibitor works. **Zone 1** indicates how long the inhibitor takes to coat the graduated cylinder. The steeper the slope, the greater the coating speed and, thus, the better the inhibitor's performance. **Zone 2** shows the moment in which the inhibitor starts to work; the corrosion rate was 0.94 mm/year with an imbalance rate of 0.22, which is a high corrosion rate according to Fontana et al. [57] with no pitting. **Zone 3** indicates the moment in which the inhibitor stops working, and in this zone, the corrosion values increased sharply. It is important to point out that the corrosion inhibitor was added only once for each test.

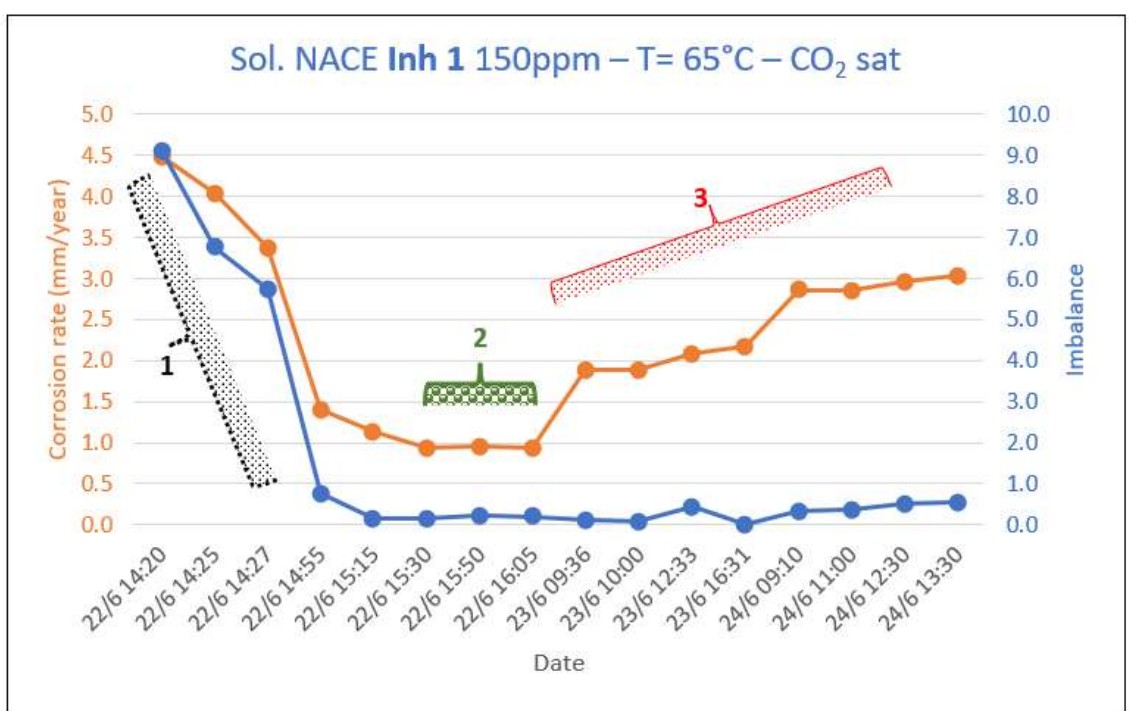

**Figure 6.** Different zones in which the inhibitor works. Zone 1: the time the inhibitor takes to coat the graduated cylinder and inhibit. Zone 2: the period during which the inhibitor works. Zone 3: the period in which the inhibitor stops working, and the corrosion rate increases.

Figure 4 corresponding to Inhibitors 2, 3, and 4, show that the **zone** differentiation was not as clear as it was for Inhibitor 1. For these cases, both the corrosion rates and the imbalance rates remained high; in addition, in the worst-case scenario, slopes are positive, as is the case for Inhibitors 3 and 4 (Figure 5), showing the presence of pitting. In the studies carried out with the rotating cylinder for Inhibitors 2, 3, and 4, efficiency was under 90% with the presence of pitting.

Observing the performance of Inhibitor 1 in Figure 4, it can be stated that the corrosion rate was 0.94 mm/year, when the inhibitor was working, whereas the data obtained in Table 3 indicate the average of all the tests, that is, the values obtained with and without the inhibitor working.

Wei Li et al. [41] precisely measured the wall shear stress under controlled laboratory conditions, under a single-phase tap-water flow in a wide range of gas–liquid flow regimes (stratified flow, slug flow, and annular mist flow). The highest wall shear stress measured was of the order of $10^2$ Pa in an atmospheric pressure flow system. Atomic force microscopy measurements indicated that the forces of the order of $10^7$ Pa would be required to remove an iron carbonate precipitate from a mild steel surface. They concluded that this is an important finding indicating that the wall shear stress typically seen in multiphase flow lines is not sufficient to damage $FeCO_3$ layers, which could lead to accelerated and localized corrosion. Although Wei Li et al., obtained important results, these are purely mechanical, and neither electrochemical phenomena nor the physical mechanisms of the solid (such as the doping of $FeCO_3$ crystals with $Na^+$, $Ca^{2+}$, ions, etc.) were considered between the protective layer and the medium (reservoir water containing high salt concentration). The studies of D. Harrop et al. [36] show a comparison of the corrosion rate for FL and RCE tests obtained under the same conditions of 1 bar $CO_2$, 50 °C, 20 Pa shear stress in the Forties formation water, with pH 5.6. Harrop concluded that this comparison was not good because the mass transfer was lower for the RCE test than for the FL under the same shear stress [37–41]. Therefore, in accordance with Harrop's perspective, we should not compare the results obtained herein for FL and RCE tests. However, the difference lies in the fact that both methods complement each other, given that the fluid rate conditions, and not those of the shear stress, were maintained for both methods. After using RCE to determine which inhibitor had a general corrosion inhibition capacity greater than 90%, under the same operating conditions, the properties of FL were reassessed under conditions more similar to what actually occurs in the field, obtaining the moment in which the protective layer was formed, the moment in which the inhibitor began to be effective, and the moment in which it started to lose its anti-corrosion capacity (Figure 6). If we combine the values of both methods, similar to how they were combined in this work, corrosion engineers are provided with more tools for corrosion control in oil fields.

## 4. Conclusions

The corrosion rates obtained using the RCE method are carried out in a very short time and are not very representative of what happens in oil and gas pipelines. By contrast, in FL tests, the times are much longer, giving more information about the behavior of the corrosion inhibitor. In flow loop tests, in addition to obtaining the corrosion rates, it is possible to discern between corrosion by wet gas and soluble gas, abrasion effects, the time the inhibitor begins to inhibit, as well as the time that the inhibitor works.

Although Inhibitor 1 offers better performance against corrosion compared with the other inhibitors tested, the protection against corrosion that it presents with the concentration used (150 ppm) is regular according to Fontana. A higher concentration of the inhibitor (e.g., 250 ppm) would substantially improve its corrosion-inhibitory properties.

**Author Contributions:** Conceptualization, G.L.B., V.A. and C.S.; Methodology, G.L.B., V.A. and C.S.; Investigation, V.A. and C.S.; Writing—review & editing, G.L.B.; Supervision, G.L.B. All authors have read and agreed to the published version of the manuscript.

**Funding:** This work was supported by TECTIONS SOUTH S.A.

**Institutional Review Board Statement:** Not applicable.

**Informed Consent Statement:** Not applicable.

**Data Availability Statement:** Not applicable.

**Conflicts of Interest:** The authors declare no conflict of interest.

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
