# Peer review of "A Combination of Laboratory Testing, RCE, and Corrosion Loop for Inhibitor Selection"

_applsci, doi:10.3390/app13074586_

Round 1

Reviewer 1 Report

The manuscript is devoted to a methodology for the evaluation of corrosion inhibitors combining RCE and FL, to select the best inhibitor performance. The manuscript is of theoretical and practical interest. However, the authors should take into account a number of questions  and remarks.

i. It is not clear how the authors calculated the corrosion rate. The formula for calculation should be given. What duration of the experiment was the corrosion rate measured at?

ii. It should be borne in mind that the corrosion rate varies with time, both in the absence and presence of an inhibitor, and the efficiency index of the inhibitor also changes with time. Therefore, it is advisable to judge the effectiveness of the inhibitor not by the value of IE, but by the corrosion rate of the metal in its presence during sufficiently long tests. An inhibitor is considered excellent if the rate of steel in its presence is 0.05 mm per year or less.

iii. According to the authors, the duration of the tests was 2-3 days, while the period during which the inhibitor 2 worked (zone 2) was only 35 minutes, while the corrosion rate was 0.94 mm/year. It is unlikely that such an inhibitor can be considered satisfactory.

iv. It is desirable that the authors of the manuscript give recommendations on how to increase the duration of successful operation of the inhibitor and reduce the steel corrosion rate in its presence to 0.05 mm per year.

Reviewer 2 Report

1. What is the optimum concentration of inhibitor which provides maximum corrosion protection efficiency

2. Include Nyquist plots for all the investigated compounds

3. Provide the fitted EIS parameters in the form of table.

4. Revise the abstract by including the key results obtained during the investigation.

5.Include the novelty of this investigation

6. Revise the conclusion part with future scope of this investigation.

7. Most of the references are outdated. Provide the recent references.

8. Revise the entire introduction part by citing the recent references.

9. SEM/EDX analysis must be included in the revised manuscript.

10. Figures are not clear. Provide clear figures.

11. Remove figure 1 as it is well known.

12. On the whole, extensive revision is required  before acceptance.

Reviewer 3 Report

This work aims to implement a methodology for evaluating corrosion inhibitors combining RCE and FL to select the inhibitor with the best performance. The topic is interesting, but the manuscript needs to be considerably improved.

The manuscript presents a weakness in its structure.

The abstract should be better structured and contain a background and motivation for the paper, a brief description of the methods, the principal results, and conclusions or interpretations.

It combines methodology with results, which makes it difficult to read.

You must update the references; less than 50% of them are from 10 years ago, and only 10% of references are from the last 5 years.

Specific comments are in the attached document

Round 2

Reviewer 2 Report

As the authors have carried out the revision as suggested by the reviewer, the manuscript may be accepted for publication after incorporating recent references in the manuscript as most of the references are very old.

Author Response

Agree. we change and update the references. See the new manuscript version 

Reviewer 3 Report

The requested changes were made to the manuscript.

Author Response

we change and update the references. See the new manuscript version 
